# Improving Policy-Constrained Kidney Exchange via Pre-Screening

**Duncan C McElfresh**
Mathematics Department
Computer Science Department
University of Maryland
College Park, MD 20742
`dmcelfre@umd.edu`

**Michael Curry**
Computer Science Department
University of Maryland
College Park, MD 20742
`curry@cs.umd.edu`

**Tuomas Sandholm**
Computer Science Department
Carnegie Mellon University
Strategy Robot, Inc.
Optimized Markets, Inc.
Strategic Machine, Inc.

**John P Dickerson**
Computer Science Department
University of Maryland
College Park, MD 20742
`john@cs.umd.edu`

## Abstract

In barter exchanges, participants swap goods with one another without exchanging money; these exchanges are often facilitated by a central clearinghouse, with the goal of maximizing the aggregate quality (or number) of swaps. Barter exchanges are subject to many forms of uncertainty–in participant preferences, the feasibility and quality of various swaps, and so on. Our work is motivated by kidney exchange, a real-world barter market in which patients in need of a kidney transplant swap their willing living donors, in order to find a better match. Modern exchanges include 2- and 3-way swaps, making the kidney exchange *clearing problem* NP-hard. Planned transplants often *fail* for a variety of reasons–if the donor organ is rejected by the recipient's medical team, or if the donor and recipient are found to be medically incompatible. Due to 2- and 3-way swaps, failed transplants can "cascade" through an exchange; one US-based exchange estimated that about $85\%$ of planned transplants failed in 2019. Many optimization-based approaches have been designed to avoid these failures; however most exchanges cannot implement these methods, due to legal and policy constraints. Instead, we consider a setting where exchanges can *query* the preferences of certain donors and recipients–asking whether they would accept a particular transplant. We characterize this as a two-stage decision problem, in which the exchange program (a) queries a small number of transplants before committing to a matching, and (b) constructs a matching according to fixed policy. We show that selecting these edges is a challenging combinatorial problem, which is non-monotonic and non-submodular, in addition to being NP-hard. We propose both a greedy heuristic and a Monte Carlo tree search, which outperforms previous approaches, using experiments on both synthetic data and real kidney exchange data from the United Network for Organ Sharing.

## 1   Introduction

We consider a multi-stage decision problem in which a decision-maker uses a fixed *policy* to solve a hard (stochastic) problem. Before using the policy, the decision-maker can first *measure* some of the uncertain problem parameters–in a sense, guiding the policy toward a better solution. Our primary motivation is kidney exchange, a process where patients in need of a kidney transplant swap their

(willing) living donors, in order to find a better match. Many government-run kidney exchanges match patients and donors using a *matching algorithm* that follows strict policy guidelines [9]; this matching algorithm is often written into law or policy, and is not easily modified. Modern kidney exchanges use both cyclical swaps and chain-like structures (initiated by an unpaired altruistic donor) [25], and identifying the max-size or max-weight set of transplants is both NP- and APX-hard [1, 7].

In kidney exchange–as in many resource allocation settings–information used by the decision-maker is subject to various forms of uncertainty. Here we are primarily concerned with uncertainty in the *feasibility* of potential transplants: if a donor is matched with a potential recipient, will the transplant actually occur? Planned transplants may *fail* for a variety of reasons: for example, medical testing may reveal that the donor and recipient are incompatible (a *positive crossmatch*); the recipient or their medical team may reject a donor organ in order to wait for a better match; or the donor may decide to donate elsewhere before the exchange is planned. Failed transplants are especially troublesome in kidney exchange, due to the cycle and chain structures used: for example, suppose that a cyclical swap is planned between three patient/donor pairs; if any one of the planned transplants fails, then none of the other transplants in that cycle can occur. Unfortunately, it is quite common for planned transplants to fail. For example, the United Network for Organ Sharing (UNOS[1]) estimates that in FY2019, about $85\%$ of their planned kidney transplants failed [18].

Various matching algorithms have been proposed that aim to mitigate transplant failures (for example, using stochastic optimization [15, 3], robust optimization [22], or conditional value at risk [6]). However, implementing these strategies would require modifying fielded matching algorithms–which in many cases would require changing law or policy. One way to avoid failures without modifying the matching algorithm is to *pre-screen* potential transplants [18, 10, 11], by communicating with the recipients' medical team and possibly using additional medical tests. Pre-screening transplants is costly, as it requires scarce time and resources. Furthermore, there are often many thousand potential transplants in any given exchange; selecting which transplants to screen is not easy.

In this paper we investigate methods for selecting a limited number of transplants to pre-screen, in order to "guide" the matching algorithm to a better outcome. We formalize this as a multistage stochastic optimization problem, and we consider both an *offline* setting (where screenings are selected all at once), and an *online* setting (where screenings are selected sequentially).

**Related Work.** While kidney exchange is known to be a hard packing problem, several algorithms exist that are scalable in practice, and are used by fielded exchanges [14, 3, 20]. Prior work has addressed potential transplant failures; our model is inspired by Dickerson et al. [15]. Pre-screening potential transplants has also been addressed in prior work ([11, 23], and § 5.1 of [13]), and our model is similar to stochastic matching and stochastic $k$-set packing [5]. However there are substantial differences between these models and ours: (a) many prior approaches assume that a large number of transplants may be pre-screened [11, 23]–on the order of one for each patient in the exchange; we assume far fewer screenings are possible; (b) prior work often assumes a *query-commit* setting– where successfully pre-screened transplants *must* be matched. Instead we assume that non-screened transplants may also be matched–which more-accurately represents the way that modern exchanges operate; (c) most prior work assumes that transplants that pass pre-screening are guaranteed to result in a transplant. In reality, transplants often fail after pre-screening, a fact reflected in our model.

One of our approaches is based on *Monte Carlo Tree Search* (MCTS), which allows efficient exploration of intractably large decision trees. While MCTS is primarily associated with Markov decision processes and game-playing [12], it has been used successfully for combinatorial optimization [16]. We use a version of MCTS, Upper Confidence Bounds for Trees (UCT), which balances exploration and exploitation by treating each tree node as a multi-armed bandit problem [4, 17].

**Our Contributions**

1. (§ 2) We formalize the *policy-constrained edge query problem*: where a decision-maker (such as a kidney exchange program) selects a set of potential edges (potential transplants) to pre-screen, prior to constructing a final packing (a set of transplants) using a fixed algorithm. This model generalizes existing models in the literature, as edge failure probabilities depend on whether or not the edge is pre-screened. Further, we allows for context-specific constraints, such as those imposed by public policy or the particular hospital or exchange.

2. (§ 3) We prove that when the decision-maker uses a max-weight packing policy (the most common choice among fielded exchanges), the edge query problem is both non-

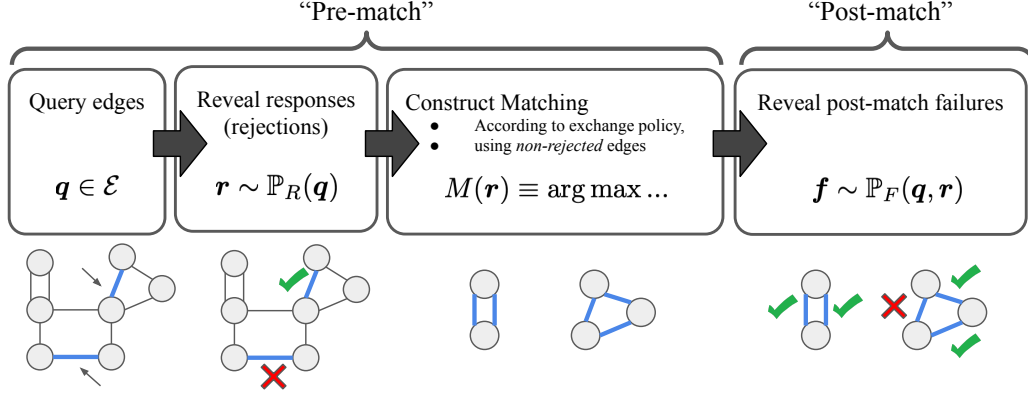

Figure 1: Single-stage edge selection: First, edges are selected to be queried, and responses revealed. Then, a final matching is constructed according to the exchange's matching policy. Finally, the post-match edge failures are revealed.

monotonic and non-submodular in the set of queried edges. Despite these worst-case findings we show that this problem is nearly monotonic for real and synthetic data, and simple algorithms perform quite well. On the other hand, when the decision-maker uses a *failure-aware* (stochastic) packing policy, the edge query problem becomes monotonic under mild assumptions.

3. (§ 4) We conduct numerical experiments on both simulated and real exchange data from the United Network for Organ Sharing (UNOS). We demonstrate that our methods substantially outperform prior approaches and a randomized baseline.

## 2 The Policy-Constrained Edge Query Problem

Kidney exchanges are represented by a graph $G = (E, V)$ where vertices $V$ represent (incompatible) patient-donor pairs, and non-directed donors (NDDs) who are willing to donate without receiving a kidney in return. Directed edges $e \in E$ between vertices represent potential transplants from the donor of one vertex to the patient of another. Edge weights represent the "utility" of an edge, and are typically set by exchange policy. Solutions to a kidney exchange problem (henceforth, *matchings*) consist of both directed *cycles* on $G$ containing only patient-donor pairs, and directed *chains* beginning with an NDD and passing through one or more pairs; see Appendix A for an example exchange graph. Each vertex may participate in only one edge in a matching–as each vertex can donate and receive at most one kidney.

Vectors are denoted in bold, and are indexed by either cycles or edges: $\boldsymbol{y}_e$ indicates the element of $\boldsymbol{y}$ corresponding to edge $e$, and $\boldsymbol{x}_c$ is the element of $\boldsymbol{x}$ corresponding to cycle $c$. Our notation uses a *cycle-chain* representation for matchings[2]: let $\mathcal{C}$ represent cycles and chains in $G$, where each cycle and chain corresponds to a list of edges; as is standard in modern exchanges, we assume that cycles and chains are limited in length. Matchings are expressed as a binary vector $\boldsymbol{x} \in \{0, 1\}^{|\mathcal{C}|}$, where $\boldsymbol{x}_c = 1$ if cycle/chain $c$ is in the matching, and $0$ otherwise. Let $\boldsymbol{w}_c$ be the weight of cycle/chain $c$ (the sum of $c$'s edge weights). Let $\mathcal{M}$ denote the set of *feasible matching*–that is, the set of vertex-disjoint cycles and chains on $G$. The total weight of a matching is simply the summed weights of all its constituent cycles and chains: $\sum_{c \in \mathcal{C}} \boldsymbol{x}_c \boldsymbol{w}_c$. We denote *sets* of edges using binary vectors, where $\boldsymbol{q} \in \{0, 1\}^{|E|}$ represents the set of all edges with $\boldsymbol{q}_e = 1$.

In the remainder of this paper we refer to pre-screening a transplant as *querying an edge*, in order to be consistent with the literature.

**Selecting Edge Queries.** Our setting consists of two phases (see Figure 1): during *pre-match*, the decision-maker selects edges to query, and each queried edge is either accepted or rejected; then the decision-maker constructs a matching using a fixed policy. During *post-match*, each match edge either fails (no transplant) or succeeds (the transplant proceeds). We consider two version of the pre-match phase: in the *single-stage* version, the decision-maker selects all queries before observing

edge responses (accept/reject); in the *multi-stage* version, one edge is selected at a time and responses are observed immediately.

Unlike most prior work, edges in our model may fail during both the pre- and post-match phase. For example, suppose the decision-maker queries an edge from a 60-year-old non-directed donor, to a 35-year-old recipient; if the recipient or their medical team rejects the elderly donor and decides to wait for a younger donor, this is a pre-match rejection. Instead suppose the edge is not queried, and it is included in the final matching; if medical screening reveals that the patient and donor are incompatible, this is a post-match failure. We refer to pre-match failures as *rejections* and post-match failures as *failures*; however we make no assumption about their cause. We represent potential failures and rejections using binary random variables: $\boldsymbol{r} \in \{0, 1\}^{|E|}$ denotes pre-match rejections, where $\boldsymbol{r}_e = 1$ if $e$ is queried and rejected, and 0 otherwise ($\boldsymbol{r}_e = 0$ for all non-queried edges). Similarly $\boldsymbol{f} \in \{0, 1\}^{|E|}$ denotes post-match failures, where $\boldsymbol{f}_e = 1$ if edge $e$ fails post-match, and 0 otherwise. We assume that the distribution of rejections $\boldsymbol{r} \sim \mathbb{P}_R(\boldsymbol{q})$ is known, and depends on $\boldsymbol{q}$; we assume the distribution of failures $\boldsymbol{f} \sim \mathbb{P}_F(\boldsymbol{q}, \boldsymbol{r})$ is known, and depends on both $\boldsymbol{q}$ and $\boldsymbol{r}$.

Rejections and failures impact the matching through the *weight* of each cycle and chain. If any cycle edge fails, then *no* transplants in the cycle can proceed; if a chain edge fails, than all edges *following* it cannot proceed.[3] Suppose we observe failures $\boldsymbol{f}$; the *final matching weight* of $c$ is

$$F(c, \boldsymbol{y}) \equiv \begin{cases} \sum_{e \in c} w_e & \text{if } \sum_{e \in c} \boldsymbol{y}_e = 0 \\ 0 & \text{if } c \text{ is a cycle and } \sum_{e \in c} \boldsymbol{y}_e > 0 \\ \sum_{e \in c'} w_e & \text{if } c \text{ is a chain, where } c' \text{ includes all edges up to the first failed edge.} \end{cases}$$

Thus the *post-match expected weight* of matching $\boldsymbol{x}$, due to both rejections $\boldsymbol{r}$ and failures $\boldsymbol{f}$, is

$$W(\boldsymbol{x}; \boldsymbol{q}, \boldsymbol{r}) \equiv \mathop{\mathbb{E}}_{\boldsymbol{f} \sim \mathbb{P}_F(\boldsymbol{q}, \boldsymbol{r})} \left[ \sum_{c \in \mathcal{C}} \boldsymbol{x}_c \, F(c, \boldsymbol{r} + \boldsymbol{f}) \right].$$

**Matching Policy** In this paper we assume that the final matching is constructed using a fixed matching policy, which uses only *non-rejected* edges; we denote this policy by $M(\boldsymbol{r})$. We focus primarily on the *max-weight* policy $M^{\texttt{MAX}}(\cdot)$, which is used by most fielded exchanges, and the *failure-aware* policy $M^{\texttt{FA}}(\cdot)$, which maximizes the expected post-match weight [15]:

$$M^{\texttt{MAX}}(\boldsymbol{r}) \in \mathop{\arg\max}_{\boldsymbol{x} \in \mathcal{M}} \sum_{c \in \mathcal{C}} \boldsymbol{x}_c \, F(c, \boldsymbol{r}) \,, \qquad M^{\texttt{FA}}(\boldsymbol{r}) \in \mathop{\arg\max}_{\boldsymbol{x} \in \mathcal{M}(\boldsymbol{r})} \mathop{\mathbb{E}}_{\boldsymbol{f} \sim \mathbb{P}_F(\boldsymbol{q}, \boldsymbol{r})} \left[ \sum_{c \in \mathcal{C}} \boldsymbol{x}_c \, F(c, \boldsymbol{r} + \boldsymbol{f}) \right] \,.$$

Evaluating this policy requires solving a kidney exchange clearing problem, which is NP-hard [1]. However, state-of-the-art method can solve realistic kidney exchange clearing problems in fractions of a second (e.g., our experiments use the PICEF method of Dickerson et al. [14]); thus, throughout this paper we treat this policy as a low- or no-cost oracle.

Next we formalize the *edge selection problem*–the main focus of this paper. We denote by $\mathcal{E}$ the set of "legal" edge subsets, subject to exchange-specific constraints; we assume that $\mathcal{E}$ is a matroid with ground set $E$. For example, the decision-maker may limit the number of queries issued to any one medical team (vertex in $G$) or transplant center (group of vertices). We aim to select an edge set $\boldsymbol{q} \in \mathcal{E}$ which maximizes the *expected weight* of the final matching. These edges are selected using only the distribution of future rejections and failures; we take a *stochastic optimization* approach, maximizing the expected outcome over this uncertainty.

**Single-Stage Setting.** The single-stage policy-constrained edge selection problem (henceforth, the *edge selection problem*) is expressed as

$$\max_{\boldsymbol{q} \in \mathcal{E}} V^S(\boldsymbol{q}), \qquad \text{with} \quad V^S(\boldsymbol{q}) \equiv \mathop{\mathbb{E}}_{\boldsymbol{r} \sim \mathbb{P}_R(\boldsymbol{q})} \left[ W\left(M(\boldsymbol{r}); \boldsymbol{q}, \boldsymbol{r}\right) \right], \tag{1}$$

where, $M(\boldsymbol{r})$ denotes the matching policy after observing rejections $\boldsymbol{r}$, and $W(\boldsymbol{x}; \boldsymbol{q}, \boldsymbol{r})$ denotes the post-match expected weight of matching $\boldsymbol{x}$. Exact evaluation of $V^S(\boldsymbol{q})$ is often intractable, as the support of $\mathbb{P}_R(\boldsymbol{q})$ grows exponentially in $|\boldsymbol{q}|$. In experiments we approximate $V^S(\boldsymbol{q})$ using sampling, and these approximations converge for a moderate number of samples (see Appendix B).

**Multistage Setting.** In the multi-stage setting, edge rejections are observed immediately after each edge is queried. The multi-stage problem is expressed as

$$\max_{\boldsymbol{q}^1 \in \mathcal{E}_1} \mathbb{E}_{\boldsymbol{r}^1 \sim \mathbb{P}_R(\boldsymbol{q}^1)} \left[ \max_{\boldsymbol{q}^2 \in \mathcal{E}_1} \mathbb{E}_{\boldsymbol{r}^2 \sim \mathbb{P}_R(\boldsymbol{q}^2)} \left[ \ldots \max_{\boldsymbol{q}^K \in \mathcal{E}_1} \mathbb{E}_{\boldsymbol{r}^K \sim \mathbb{P}_R(\boldsymbol{q}^K)} \left[ W\left(M(\boldsymbol{r}); \boldsymbol{q}, \boldsymbol{r}\right) \right] \right] \ldots \right], \quad (2)$$

where $\boldsymbol{q} \equiv \sum_{i=1}^K \boldsymbol{q}^i$ denotes all queried edges, $\boldsymbol{r} \equiv \sum_{i=1}^K \boldsymbol{r}^i$ denotes all rejections, and $\mathcal{E}_1 \subseteq \mathcal{E}$ be denotes the legal edge subsets containing only one edge. First, we observe that Problems 1 and 2 require evaluating a matching policy $M(\boldsymbol{r})$. In the case of kidney exchange, evaluating both the max-weight policy $M^{\texttt{MAX}}(\cdot)$ and the failure-aware policy $M^{\texttt{FA}}(\cdot)$ require solving NP-hard problems; thus Problems 1 and 2 are at least NP-hard as well.

However, regardless of matching policy, the question whether *edge selection* is is hard. We observe that while these problems are difficult in principle, experiments (§ 4) show that they are easy in practice. Proofs of the following propositions can be found in Appendix D.

**Proposition 2.1.** *With matching policy $M^{\texttt{FA}}(\cdot)$, the objective of Problem 1 is non-monotonic in the number of queried edges, even with independent edge distributions.*

In other words, querying additional edges can sometimes lead to a *worse* outcome. This is somewhat counter-intuitive; one might think that providing additional information to the matching policy would strictly improve the outcome. This is a worst-case result–and in fact our experiments demonstrate that querying edges almost always leads to a better final matching weight.

**Proposition 2.2.** *With matching policy $M^{\texttt{MAX}}(\cdot)$, the objective of Problem 1 is non-submodular in the set of queried edges.*

In other words, certain edges are *complementary* to each other–and querying complementary edges simultaneously can yield a greater improvement than querying them separately. Taken together, these propositions indicate that single-stage edge selection with matching policy $M^{\texttt{MAX}}(\cdot)$ is a challenging combinatorial optimization problem. On the other hand, using the failure-aware matching policy $M^{\texttt{FA}}(\cdot)$ allows us to avoid some of these issues.

**Proposition 2.3.** *With matching policy $M^{\texttt{FA}}(\cdot)$, and if all edges are independent, the objective of Problem 1 is monotonic in the set of queried edges.*

While Propositions 2.1 and 2.2 state that single-stage edge selection is challenging in the worst case, our computational results suggest that these problems are often easier on realistic exchanges.

## 3    Solving the Policy-Constrained Edge Query Problem

First we propose an exhaustive tree search which returns an optimal solution to Problem 1 given enough time. Building on this, we propose a Monte Carlo Tree Search algorithm and a simple greedy algorithm. Our multi-stage approaches are very similar to these, and can be found in Appendix E.

Our optimal exhaustive search uses a *search tree* where each tree node corresponds to an edge subset in $\boldsymbol{q} \in \mathcal{E}$. The children of node $\boldsymbol{q}$ correspond to any $\boldsymbol{q}' \in \mathcal{E}$ which are equivalent to the parent $\boldsymbol{q}$, but include one additional edge: $C(\boldsymbol{q}) \equiv \{(\boldsymbol{q} + \boldsymbol{q}') \, \forall \boldsymbol{q}' \in \mathcal{E} : |\boldsymbol{q}'| = 1 \mid (\boldsymbol{q} + \boldsymbol{q}') \in \mathcal{E}\}$ . We say that edge sets (or tree nodes) containing $L$ edges are on the $L^{th}$ *level* of the tree. We refer to nodes with no children as *leaf nodes*. Unlike other tree search settings, the optimal solution to Problem 1 may be at *any* node of the tree, not only leaf nodes; this is a consequence of non-monotonicity (see Proposition 2.1). The tree defined by root node $\boldsymbol{q} = \boldsymbol{0}$ and child function $C(\boldsymbol{q})$ contains all legal edge subsets in $\mathcal{E}$, when $\mathcal{E}$ is a matroid. Thus, *any* exhaustive tree search algorithm (such as depth-first search) will identify an optimal solution, given enough time and memory.

Of course exhaustive search is only tractable if $\mathcal{E}$ is small. Consider the class of *budgeted* edge sets $\mathcal{E}(\Gamma)$ used in our experiments: $\mathcal{E}(\Gamma) \equiv \{\boldsymbol{q} \in \{0,1\}^{|E|} \mid |\boldsymbol{q}| \leq \Gamma\}$ (edge sets containing at most $\Gamma$ edges). The number of edge sets in $\mathcal{E}(\Gamma)$ grows roughly exponentially in $\Gamma$ and $|E|$, and is impossible to enumerate even for small graphs. Suppose a graph has 50 edges and we have an edge budget of five: there are over two million edge sets in $\mathcal{E}(5)$. Even small exchange graphs can have thousands of edges, and thus $\mathcal{E}(\Gamma)$ cannot be enumerated. Therefore, we propose search-based approach.

**Monte Carlo Tree Search for Edge Selection (**MCTS**):** We propose a tree-search algorithm for single-stage edge selection, MCTS, based on Monte Carlo Tree Search (MCTS), with the Upper Confidence for Trees (UCT) algorithm [17]. Our approach keeps track of a *value* (the objective value of Problem 1) and a UCB value estimate for each node, and these values are updated during sampling.

The formula used to estimate a node's UCB value is

$$\frac{\frac{U}{N} - V^{min}}{V^{max} - V^{min}} + \sqrt{N^P/N}$$

where $U$ is the "UCB value estimate" calculated by `MCTS`, $N$ is the number of visits to the node, $N^P$ is the number of visits to the node's parent, and $V^{max}$ and $V^{min}$ are the largest and smallest node values encountered during search.

When the set of tree nodes is too large to enumerate UCT can use a huge amount of memory–by storing values for each visited node. To limit both memory use and runtime, we incrementally search the tree from a temporary root node. Beginning from the root (the the empty edge set), we use UCB sampling on the next $L$ levels of nodes–where $L$ is a small fixed integer. After a fixed time limit, sampling stops and we set the *new* root node to the current root's best child according to its UCB estimate–using the method of [17]. This process repeats until we reach the final level of the search tree. Algorithm 1 gives a pseudocode description of `MCTS`, which uses Algorithm 2 as a submethod. While often successful, MCTS requires extensive training and parameter tuning. As a simpler alternative, we propose a greedy algorithm.

**Single-Stage Greedy Algorithm: `Greedy`.** Like `MCTS`, our greedy algorithm (`Greedy`) begins with the empty edge set as the root node, and iteratively searches deeper levels of the tree. However unlike `MCTS`, `Greedy` simply selects the child node with the greatest objective value in Problem 1–that is, *greedily* improving the objective value; see Appendix E for a pseudocode description.

| **ALGORITHM 1:** `MCTS`: Tree Search for Single-Stage Edge Selection | **ALGORITHM 2:** `Sample`: Sampling function used by `MCTS` |
|---|---|
| (input) $K$: maximum size of any legal edge set <br> (input) $T$: time limit per level <br> (input) $L$: number of look-ahead levels <br><br> $q^R \leftarrow \mathbf{0}$    root node (no edges) <br> $q^* \leftarrow \mathbf{0}$    the best visited node <br> $V^* \leftarrow$ objective value of $q^*$ <br> **for** $N = 1, \ldots, K$ **do** <br>      $M \leftarrow \min\{N + L, K\}$ <br>      $Q \leftarrow$ all nodes in levels $N$ to $M$ <br>      $U[q] \leftarrow 0 \ \forall q \in Q$    UCB value estimate <br>      $V[q] \leftarrow 0 \ \forall q \in Q$    objective value <br>      $N[q] \leftarrow 0 \ \forall q \in Q$    number of visits <br>      **while** *less than time $T$ has passed* **do** <br>          `Sample`$(q^R, M)$ <br>      $q^R \leftarrow \arg\max_{q \in C(q^R)} U[q]$ <br>      Delete $U[\cdot]$, $V[\cdot]$, and $N[\cdot]$ <br> **return** $q^*$ | (input) $q$, $M$ <br><br> $N[q] \leftarrow N[q] + 1$ <br> $V[q] \leftarrow$ objective of edge set $q$ in Problem 1 <br> **if** $V[q] > V^*$ **then** <br>      $q^* \leftarrow q$, $V^* \leftarrow V[q]$ <br> **if** *$q$ has no children* **then** <br>      **return** $V[q]$ <br> **if** *$q$ has children* **then** <br>      **if** $|q| < M$ **then** <br>          $q' \leftarrow \arg\max_{q \in C(q^R)} U[q] + \text{UCB}[q]$ <br>          $U[q] \leftarrow U[q] + $ `Sample`$(q', M)$ <br>      **else** <br>          $q' \leftarrow$ a random descendent of $q$ at any level <br>          $V' \leftarrow$ objective value of $q'$ in Problem 1 <br>          **if** $V' > V^*$ **then** <br>              $q^* \leftarrow q'$, $V^* \leftarrow V'$ <br>      $U[q] \leftarrow U[q] + V'$ |

**Runtime.** Our methods rely on an "oracle" to solve the NP-hard kidney exchange matching problem; while state-of-the-art methods solve real-sized instances of these problems in fractions of a second, there is no guaranteed bound for absolute runtime. Instead, we can report the *number of calls* to this oracle for each method as a measure of complexity. Both benchmark methods (max-weight matching and failure-aware [15]) as well as `IIAB` [11] use exactly one oracle call; i.e., they are $O(1)$. Both `Greedy` and `MCTS` use a fixed number of samples ($M$) to evaluate the objective of an edge set. `Greedy` evaluates the objective of an edge set exactly $\Gamma$ times; thus, `Greedy` is $O(M \cdot \Gamma)$. Finally, `MCTS` can in theory visit all potential edge sets of size at most $\Gamma$ (i.e., an exhaustive search), which is $O(M \cdot \sum_{\gamma=1}^{\Gamma} \binom{|E|}{\gamma})$. Since this version of `MCTS` is intractable in both runtime and memory, Algorithm 1 imposes reasonable limits on our implementation.

## 4 Computational Experiments

We conduct a series of computational experiments using both synthetic data, and real kidney exchange data from UNOS; all code for these experiments is available online.[4] In these experiments, "legal" edge sets are the budgeted edge sets defined as $\mathcal{E}(\Gamma) \equiv \{q \in \{0,1\}^{|E|} \mid |q| \leq \Gamma\}$.

In Sections 4.1 and 4.2 we present results in the single- and multi-stage edge selection settings, respectively. We use two types of data for these experiments:

**Real Data.** We use exchange graphs from the United Network for Organ Sharing (UNOS), representing UNOS match runs between 2010 and 2019. Some of these exchange graphs only have the trivial matching (no cycles or chains), or they have only one non-trivial matching. We ignore these graphs because the matching policy is a "constant" function (to return the one feasible matching) and edge queries cannot change the outcome. Removing these, we are left with 324 UNOS exchange graphs.

**Synthetic Data.** We generate random kidney exchange graphs based on directed Erdős-Rényi graphs defined using parameters $N$ and $p$: let $V$ be a fixed set of $N$ vertices; for each pair of vertices $(V_1, V_2)$ there is an edge from $V_1$ to $V_2$ with probability $p$, and an edge from $V_2$ to $V_1$ with probability $p$ (independent of the edge from $V_1$ to $V_2$). Any vertices with no incoming edges are considered NDDs.

In these experiments edge rejections and failures are independently distributed for each edge $e$; let $P_R$ be the rejection probability, $P_Q$ is the post-match success probability if $e$ is queried/accepted, and $P_N$ is the success probability if $e$ is not queried. To simulate edge rejections and failures we use two synthetic edge distributions: *Simple* and *KPD*. In the *Simple* distribution, $P_R = 0.5$, $P_Q = 1$, and $P_N = 0.5$ for all edges. The *KPD* distribution is inspired by the fielded exchange setting from which we draw our real underlying compatibility graphs. According to UNOS, about 34% of all edges are rejected by a donor or recipient pre-match [18]; we draw $P_R$ uniformly from $U(0.25, 0.43)$ for each edge. Edges ending in highly-sensitized patients (who are often less healthy and more likely to be incompatible) are considered high-risk; for these edges we draw $P_Q$ from $U(0.2, 0.5)$ and $P_N$ from $U(0.0, 0.2)$. For other edges we draw $P_Q$ from $U(0.9, 1.0)$ and $P_N$ from $U(0.8, 0.9)$.

## 4.1 Single-Stage Edge Selection Experiments

In this section we compare against the baseline of a max-weight matching *without* edge queries (using policy $M^{\texttt{MAX}}(\cdot)$). Many fielded kidney exchanges use a variant of this matching policy, so by comparing against this baseline we are illustrating the impact of edge queries on the state-of-the-art matching policies used in many real exchanges. Let $V_X$ be the objective[5] of Problem 1 achieved by method $X$, we calculate $\Delta^{\texttt{MAX}}$ (the relative difference from baseline) as $\Delta^{\texttt{MAX}} \equiv (V_X - V^S(\mathbf{0}))/V^S(\mathbf{0})$. A value of $\Delta^{\texttt{MAX}} = 0$ means that method $X$ did not improve over the baseline, a value of $\Delta^{\texttt{MAX}} = 1$ means that $X$ achieved an objective 100% greater than the baseline, and so on. Furthermore a value of $\Delta^{\texttt{MAX}} > 0$ means that method $X$ *increases* the objective by querying edges, while $\Delta^{\texttt{MAX}} < 0$ means that method $X$ decreases the objective by querying edges.

**Result: `Greedy` is essentially Optimal with small random graphs.** First we investigate the *difficulty* of edge selection. Using random graphs, we compare `Greedy` to the *optimal* solution to Problem 1, found by exhaustive search (`OPT`). We generate three sets of 100 random graphs with $N = 50$, 75, and 100 vertices, and each with $p = 0.01$. For all graphs we run both `OPT` and `Greedy` with edge budget 3; we calculate the *optimality gap* of `Greedy` as $\%\texttt{OPT} \equiv 100 \times (V_{\texttt{OPT}} - V_{\texttt{Greedy}})/V_{\texttt{OPT}}$, where $V_X$ denotes the objective achieved by method $X$. ($V_{\texttt{OPT}} > 0$ in all graphs used in these experiments.) If $\%\texttt{OPT} = 0$ then `Greedy` returns an optimal solution, and $\%\texttt{OPT} > 0$ means that `Greedy` is not optimal. Table 1 (left) shows the number of random graphs binned by $\%\texttt{OPT}$, as well as the maximum $\%\texttt{OPT}$ over all graphs. For each $N$, `Greedy` returns an optimal solution for at least 90 of the 100 graphs; the *maximum* $\%\texttt{OPT}$ over all graphs is 2.8.

In other words, `Greedy` always returns an *optimal* or nearly-optimal set of edges to query for small random graphs. This is somewhat unexpected, since the edge selection problem is both non-monotone and non-submodular (see Section 2).

**Result: `Greedy` is essentially monotonic with UNOS graphs.** We test `Greedy` on real UNOS graphs, using maximum budget 100. Figure 2a shows the median $\Delta^{\texttt{MAX}}$ over all UNOS graphs, with shading between the $10^{th}$ and $90^{th}$ percentiles. Larger edge budgets almost never decrease the objective achieved by `Greedy`, and `Greedy` *never* produces a worse outcome than the baseline. Thus–in our setting–single-stage edge selection is effectively monotonic in our setting, and `Greedy` is an effective method.

**Result: `MCTS` and `Greedy` are nearly equivalent with UNOS graphs.** We compare all methods on UNOS graphs, using smaller, more-realistic edge budgets from 1 to 10. For `MCTS` we use a 1-hour time limit per edge ($\Gamma$ hours total). Figures 2b and 2d compare $\Delta^{\texttt{MAX}}$ for `MCTS`, `Greedy`, and random

[†] We use an approximation of Fail-Aware for the *KPD* dist.; *true* Fail-Aware should always have $\Delta^{\texttt{MAX}} > 0$.

Table 1: Left: Optimality gap for `Greedy`, over 100 random graphs with $p = 0.01$ and various $N$, with edge budget $\Gamma = 3$; bottom row shows the maximum value of %OPT over all graphs. Right: Single-stage results on UNOS graphs using the variable IIAB edge budget (top rows), and the failure-aware method (bottom row). Columns $P_X$ indicates the $X^{th}$ percentile of $\Delta^{\texttt{MAX}}$ over all UNOS graphs.

| %OPT | Num. Graphs (out of 100) | | |
|---|---|---|---|
| | $N = 50$ | $N = 75$ | $N = 100$ |
| $[0, 0.1]$ | 93 | 93 | 90 |
| $(0, 1]$ | 5 | 4 | 9 |
| $(1, 2]$ | 1 | 3 | 1 |
| $(2, 100]$ | 1 | 0 | 0 |
| Max %OPT | 2.8 | 1.5 | 1.0 |

| Method | *Simple* edge dist. | | | *KPD* edge dist. | | |
|---|---|---|---|---|---|---|
| | $P_{10}$ | $P_{50}$ | $P_{90}$ | $P_{10}$ | $P_{50}$ | $P_{90}$ |
| MCTS | 0.40 | 0.67 | 1.11 | 0.05 | 0.45 | 3.44 |
| Greedy | 0.47 | 0.64 | 1.00 | 0.02 | 0.47 | 3.44 |
| Random | 0.00 | 0.10 | 0.46 | −0.11 | 0.00 | 0.63 |
| IIAB | 0.21 | 0.45 | 0.89 | −0.27 | 0.12 | 2.24 |
| Fail-Aware | 0.00 | 0.09 | 0.23 | −0.27$^\dagger$ | 0.00$^\dagger$ | 2.17$^\dagger$ |

edge selection, for the *Simple* and *KPD* edge distributions, respectively. We draw two conclusions from these results: (1) MCTS and Greedy produce almost identical results, further suggesting that Greedy is nearly optimal in our setting; (2) in our setting, edge selection is *effectively* monotonic, as $\Delta^{\texttt{MAX}}$ almost never decreases. However Figure 2d gives an example of non-monotonicity for both Greedy and Random: in some cases, querying edges can lead to a *worse* outcome than querying no edges.

**Result: Both MCTS and Greedy outperform benchmarks from the literature.** We also compare against two state-of-the-art approaches: the edge selection approach of [11] (IIAB), which uses a *variable* edge budget that depends on the graph structure; and and the failure-aware matching policy of [15] (Fail-Aware[6]), which does not query edges To our knowledge, IIAB is the only edge selection method in the literature. We compare against the Fail-Aware method because it is a state-of-the-art kidney exchange matching policy which aims to maximize the expected matching weight, under a similar edge failure model to ours; we compare against this approach to further illustrate the utility of querying edges.

Table 1 (right) shows a comparison of all edge-selection methods–each using the variable edge budget of IIAB; the bottom row shows results for Fail-Aware. Both MCTS and Greedy achieve greater $\Delta^{\texttt{MAX}}$ (in distribution) than both benchmark methods. This is expected in both cases: IIAB uses a heuristic to select edges to query, which does not consider the final matching weight—the objective of our edge selection problem; on the other hand, both MCTS and Greedy are designed to maximize this objective. We do not expect Fail-Aware to out-perform any edge selection methods, since Fail-Aware does not have access to information revealed after edge queries.

It is notable that Greedy performs better than MCTS (in distribution). This likely means that MCTS is *under-trained*—that the time and memory limits used in our implementation are too restrictive; alternatively, this indicates that Greedy is simply very effective in our setting.

### 4.2 Multi-Stage Edge Selection Experiments on UNOS Graphs

We run initial multi-stage edge selection experiments on all UNOS graphs with the *Simple* edge distribution. For each graph we test our multi-stage variants of MCTS and Greedy, and compare with a baseline of random edge selection; as before, MCTS uses a 1-hour training time per level. It is substantially harder to evaluate the multi-stage objective, as each edge edge-selection method changes depending on rejections observed in prior stages. Similarly, the MCTS search tree is orders of magnitude larger in the multi-stage setting: each node in tree corresponds to both an edge set *and* a rejection scenario (see Appendix E).

In these initial experiments we evaluate each method on 10 edge rejections *realizations* (only a small subset). We estimate $\Delta^{\texttt{MAX}}$ for each method and each graph by averaging the final matching weight over all realizations. Figure 2c shows the results of these experiments.

These initial multi-stage results are quite similar to our single-stage results. However it is notable that the objective value in the multi-stage setting is somewhat higher than in the single-stage setting– even using the simple method Greedy. Further, this suggests that more can be gained by developing a more sophisticated multi-stage edge selection policy. We leave this for future work.

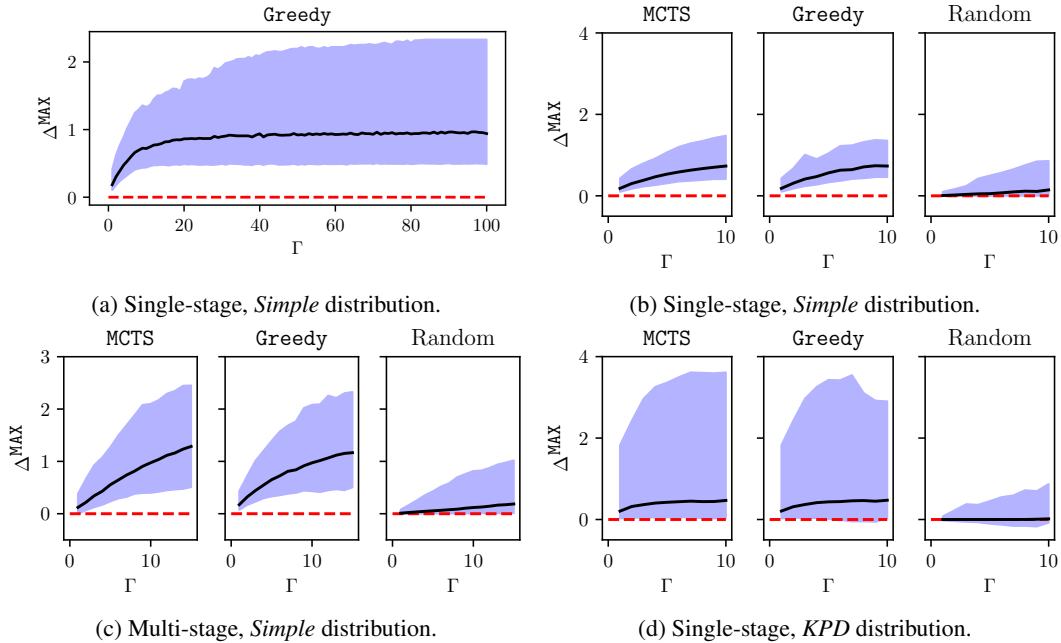

(a) Single-stage, *Simple* distribution.

(b) Single-stage, *Simple* distribution.

(c) Multi-stage, *Simple* distribution.

(d) Single-stage, *KPD* distribution.

Figure 2: Results for UNOS graphs. Right: edge budget up to 10 for the *Simple* distribution (top) and the *KPD* distribution (bottom). Top-left: Greedy with edge budget up to 100, for the simple distribution. Bottom-left: multi-stage methods using the *Simple* distribution. In all plots, a solid line indicates median $\Delta^{\text{MAX}}$ over all UNOS graphs, and shading is between the $10^{th}$ and $90^{th}$ percentiles; a dotted line indicates the baseline.

## 5   Conclusions and Future Research Directions

Many planned kidney exchange transplants *fail* for a variety of reasons; these failures greatly reduce the number of transplants that an exchange can facilitate, and increase the waiting time for many patients in need of a kidney. Avoiding transplant failures is a challenge, as exchanges are often constrained by policy and law in how they match patients and donors. We consider a setting where exchanges can *pre-screen* certain transplants, while still matching patients and donors using a fixed policy. We formalize a multi-stage optimization problem based on realistic assumptions about how transplants fail, and how exchanges match patients and donors; we emphasize that these important assumptions are not included in prior work. While this problem is challenging in theory, we show that it is much easier in practice–with computational experiments using both synthetic data and real data from the United Network for Organ Sharing. In experiments, we find that pre-screening even a small number of potential transplants (around 10) significantly increases the overall quality of the final match–by more than 100% of the original match weight.

Our initial study of the pre-screening problem suggests several areas for future work. First we assume that the distribution of transplant failures is known, when in reality only rough approximations of these distributions are available. Second, we assume that exchange participants (donors, recipients, hospitals) are not strategic. In reality, strategic behavior plays a substantial role in real exchanges [2]; we expect that participants might behave strategically when responding to pre-screening requests. Third, our model does not account for equitable treatment of different patients [21]. For example, it may be the case that pre-screening a transplant decreases the likelihood of the transplant being matched. That might disproportionately impact highly-sensitized patients, which are both sicker and more difficult to match than other patients.

## Broader Impact

This work lives within the broader context of kidney exchange research. For clarity, we separate our broader impacts into two sections: first we discuss the impact of kidney exchange in general; then we discuss our work in particular, within the context of kidney exchange research and practice.

**Impacts of Kidney Exchange**    Patients with end-stage renal disease have only two options: receive a transplant, or undergo dialysis once every few days, for the rest of their lives. In many countries (including the US), these patients register for a deceased donor waiting list–and it can be months or years before they receive a transplant. Many of these patients have a friend or relative willing to donate a kidney, however many patients are incompatible with their corresponding donor. Kidney exchange allows patients to "swap" their incompatible donor, in order to find a *higher-quality* match, *more quickly* than a waiting list. Transplants allow patients a higher quality of life, and cost far less, than lifelong dialysis. About $10\%$ of kidney transplants in the US are facilitated by an exchange.

Finding the "most efficient" matching of kidney donors to patients is a (computationally) hard problem, which cannot be solved by hand in most cases. For this reason many fielded exchanges use algorithms to quickly find an efficient matching of patients and donors. Many researchers study kidney exchange from an algorithmic perspective, often with the goal of improving the number or quality of transplants facilitated by exchanges. Indeed, this is the purpose of our paper.

**Impacts of Our Work**    In this paper we investigate the impact of pre-screening certain potential transplants (edge) in an exchange, prior to constructing the final patient-donor matching. To our knowledge, some modern fielded exchanges pre-screen potential transplants in an ad-hoc manner; meaning they do not consider the impacts of pre-screening on the final matching. We propose methods to estimate the importance of pre-screening each edge, as measured by the change in the overall number and quality of matched transplants.[7] Importantly, our methods do not require a change in matching policy; instead, they indicate to policymakers which potential transplants are important to pre-screen, and which are not. The impacts of our contributions are summarized below:

**Some potential transplants cannot be matched**, because they cannot participate in a "legal" cyclical or chain-like swap (according to the exchange matching policy). Accordingly, there is no "value" gained by pre-screening these transplants; our methods will identify these potential transplants, and will recommend that they not be pre-screened. Pre-screening requires doctors to spend valuable time reviewing potential donors; removing these unmatchable transplants from pre-screening will allow doctors to focus only on transplants that are relevant to the current exchange pool.

**Some transplants are more important to pre-screen than others**, and our methods help identify which are most important for the final matching. We estimate the value pre-screening of each transplant by *simulating* the exchange matching policy in the case that the pre-screened edge is pre-accepted, and in the case that it is pre-refused.

**To estimate the value of pre-screening each transplant, we need to know (a) the likelihood that each transplant is pre-accepted and pre-refused, and (b) the likelihood that each planned transplant fails for any reason, after being matched.** These likelihoods are used as input to our methods, and they can influence the estimated value of pre-screening different transplants. Importantly, it may not be desirable to calculate these likelihoods for each potential transplant (e.g., using data from the past). For example if a patient is especially sick, we may estimate that any potential transplant involving this patient is very likely to fail prior to transplantation (e.g., because the patient is to ill to undergo an operation). In this case, our methods may estimate that all potential transplants involving this patient have very low "value", and therefore recommend that these transplants should not be pre-screened. One way to avoid this issue is to use the same likelihood estimates for all transplants.

**To estimate the impact of our methods (and how they depend on the assumed likelihoods, see above), we recommend using extensive modeling of different pre-screening scenarios before deploying our methods in a fielded exchange.** This is important for several reasons: first, exchange programs cannot always *require* that doctors pre-screen potential transplants prior to matching. Since we cannot be sure which transplants will be pre-screened and which will not, simulations should be run to evaluate each possible scenario. Second, theoretical analysis shows that pre-screening transplants can—in the worst case—negatively impact the final outcome. While this worst-case

outcome is possible, our computational experiments show that it is very unlikely; this can be addressed further with mode experiments tailored to a particular exchange program.

## Acknowledgments

We thank Ruthanne Leishman and Morgan Stuart at UNOS for very helpful early-stage discussions, feedback on our general approach, clarifications regarding data, and knowledge of the intricacies of running a fielded exchange. Curry, Dickerson, and McElfresh were supported in part by NSF CAREER Award IIS-1846237, DARPA GARD, DARPA SI3-CMD #S4761, DoD WHS Award #HQ003420F0035, NIH R01 Award NLM-013039-01, and a Google Faculty Research Award. Sandholm was supported in part by the National Science Foundation under grants IIS-1718457, IIS-1617590, IIS-1901403, and CCF-1733556, and the ARO under award W911NF-17-1-0082.

## Footnotes

[1]UNOS is the organization tasked with overseeing organ transplantation in the US: `https://unos.org/`.

[2]Our experiments use the position-indexed formulation, which is more compact and equivalent [14].

[3]This assumes that chains can be *partially* executed: for example, suppose that the $4^{th}$ edge in a 10-edge chain fails; the first three edges can still be matched, and the post-failure chain weight sums only these three edges. Not all fielded exchanges use this policy: some exchanges cancel the entire chain if one of its edges fails.

[4] `https://github.com/duncanmcelfresh/kpd-edge-query`

[5] All objective values are estimated using up to 1000 sampled rejection scenarios (see Appendix B), as it is intractable to evaluate the exact objective of large edge sets.

[6]For the *KPD* distribution we use an approximation of Fail-Aware, which assumes a uniform edge failure probability.

[7]Quality and quantity of transplants is measured by transplant weight, a numerical representation of transplant quality (e.g., see UNOS/OPTN Policy 13 regarding KPD prioritization points `https://optn.transplant.hrsa.gov/media/1200/optn_policies.pdf`).

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
