[Supplementary Material]



Figure 3: Sample exchange graph with a 3-chain (dashed edges) and two 2-cycles (solid edges). The NDD is denoted by $n$, and each patient (and associated donor) is denoted by $p_i$ ($d_i$). If edge $e_1$ is not queried, or queried and *accepted*, then the chain may be included in the final matching. However if edge $e_A$ is queried and *rejected*, then only the 2-cycles may be included in the final matching.

## A    Kidney Exchange and Edge Failures

**Brief history.** Rapaport [25] proposed the initial idea for kidney exchange, while the first organized kidney exchange, the New England Paired Kidney Exchange (NEPKE), started in 2003–04 [27, 28, 29]. NEPKE has since ceased to operate; at the point of cessation, its pool of patients and donors was merged into the United Network for Organ Sharing (UNOS) exchange in late 2010. That exchange now contains over 60% of transplant centers in the US, and performs matching runs via a purely algorithmic approach (as we discuss in Sections 1 and 2, and in much greater depth by UNOS [30], which is mandated to transparently and publicly reveal its matching process).

There are also two large private kidney exchanges in the US, the National Kidney Registry (NKR) and the Alliance for Paired Donation (APD). They typically only work with large transplant centers. NKR makes their matching decisions manually and greatly prefers matching incrementally through chains. APD makes their decisions through a combination of algorithmic and manual decision making. There are also several smaller private kidney exchanges in the US. They typically only involve one or a couple of transplant centers. These include an exchange at Johns Hopkins University, a single-center exchange at the Methodist Specialty and Transplant Hospital in San Antonio, and a single-center exchange at Barnes-Jewish Hospital affiliated with the Washington University in St. Louis. Largely, these exchanges also make their matching decisions via a combined algorithmic and manual process. These exchanges compete in a variety of ways (e.g., by allowing patient-donor pairs to register in multiple exchange programs); this competition can lead to loss in efficiency [2] as well as sub-optimal changes to individual exchanges' matching polices [20].

There are now established kidney exchanges in the UK [21], Italy, Germany, Netherlands, Canada, England, Portugal, Israel, and many other countries. European countries are also explicitly exploring connecting their individual exchanges together in various ways [8].

**Edge failures.** The dilemma of edge failures is illustrated in the example exchange graph shown in Figure 3. This exchange consists of a 3-chain (dashed edges) and two 2-cycles (solid edges). Suppose the decision-maker queries edge $e_A$: if $e_A$ is accepted, then the chain from the NDD ($n$) through pairs $(d_1, p_1)$, $(d_2, p_2)$, and $(d_3, p_3)$, i.e., the dashed edges, can be included in the matching. However if $e_A$ is queried and rejected, then the NDD cannot initiate the chain, and only the cycles may be matched. In our model, if $e_A$ is not queried then it may still be matched.

## B    Estimating The Objective of Problem 1

The objective of the single-stage edge selection problem requires evaluating all rejection scenarios $\boldsymbol{r} \sim \mathbb{P}_R(\boldsymbol{q})$, and the support of this distribution grows exponentially in the number of edges $|\boldsymbol{q}|$. In computational experiments, to estimate the objective of Problem 1 we sample up to 1000 scenarios from $\mathbb{P}_R(\boldsymbol{q})$. More explicitly: we *exactly* evaluate the objective of edge sets with fewer than 10 edges; for larger edge sets, we sample the objective using 1000 draws from $\mathbb{P}_R(\boldsymbol{q})$.

Using bootstrapping experiments we demonstrate that our sampling approach is sufficient to accurately estimate the true objective, even for large edge sets. For 152 UNOS graphs, we computed edge sets by running `Greedy` with edge budgets ranging from 1 to 100. For each edge set, we then sample a subset of $N \in \{10, 30, 50, 100, 1000\}$ rejection scenarios, with replacement, from the set of all sampled edge outcomes. For each edge set and choice of $N$ we repeat 200 times and calculate the sample mean for each replication. We then compute the standard deviations of these bootstrap

| Edge budgets | $N = 10$ | $N = 30$ | $N = 50$ | $N = 100$ | $N = 1000$ |
|---|---|---|---|---|---|
| 1-10 | 0.10 | 0.06 | 0.04 | 0.03 | 0.01 |
| 11-20 | 0.12 | 0.07 | 0.05 | 0.04 | 0.01 |
| 21-30 | 0.13 | 0.08 | 0.06 | 0.04 | 0.01 |
| 31-40 | 0.14 | 0.08 | 0.06 | 0.04 | 0.01 |
| 41-50 | 0.14 | 0.08 | 0.06 | 0.04 | 0.01 |
| 51-60 | 0.15 | 0.08 | 0.07 | 0.05 | 0.01 |
| 61-70 | 0.15 | 0.09 | 0.07 | 0.05 | 0.02 |
| 71-80 | 0.16 | 0.09 | 0.07 | 0.05 | 0.02 |
| 81-90 | 0.17 | 0.10 | 0.08 | 0.05 | 0.02 |
| 91-100 | 0.18 | 0.10 | 0.08 | 0.06 | 0.02 |

Table 2: Median normalized standard deviation of the bootstrap mean, over 200 bootstrap samples for each sample size $N$, binned by edge budget.

Table 3: Single-stage results on random graphs with the *Simple* edge distribution, using the variable IIAB edge budget (top rows), and the failure-aware method (bottom row). Columns $P_X$ indicates the $X^{th}$ percentile of $\Delta^{\texttt{MAX}}$ over all 30 random graphs, for graphs with $N = 50, 75$, and 100 vertices.

| | $N = 50$ | | | $N = 75$ | | | $N = 100$ | | |
|---|---|---|---|---|---|---|---|---|---|
| **Method** | $P_{10}$ | $P_{50}$ | $P_{90}$ | $P_{10}$ | $P_{50}$ | $P_{90}$ | $P_{10}$ | $P_{50}$ | $P_{90}$ |
| MCTS | 0.22 | 0.30 | 0.38 | 0.11 | 0.33 | 0.46 | 0.23 | 0.33 | 0.38 |
| Greedy | 0.21 | 0.30 | 0.38 | 0.12 | 0.32 | 0.48 | 0.27 | 0.39 | 0.43 |
| Random | 0.12 | 0.19 | 0.23 | 0.10 | 0.19 | 0.28 | 0.12 | 0.19 | 0.23 |
| IIAB | 0.07 | 0.24 | 0.34 | 0.11 | 0.22 | 0.41 | 0.07 | 0.24 | 0.34 |
| Fail-Aware | 0.00 | 0.02 | 0.10 | 0.00 | 0.06 | 0.18 | 0.00 | 0.02 | 0.10 |

sample means to estimate the variance due to sampling. For each $N$, we calculate the mean sample standard deviation, normalized by the sample mean. Table 2 shows the median normalized standard deviation for all experiments under each $N$, with edge budgets aggregated into 10 bins. We find that with $N = 1000$ samples, the standard deviation was on average only about 2% of the overall mean value, even for large edge budgets.

## C  Additional Computational Results

First we show results for both single-stage and multi-stage edge selection on random graphs (see § 4 for a description of these graphs). For $N = 50, 75$, and 100, we generate 30 random graphs with $N$ vertices and $p = 0.01$. For each graph we run single-stage experiments with $\Gamma = 1, \ldots, 10$ and multi-stage experiments with $\Gamma = 1, \ldots, 15$. Unlike experiments on UNOS graphs we use a time limit of 20 minutes per edge; all other parameters are the same. Figure 4a and 4b show single-stage and multi-stage results for all random graphs, respectively. Table 3 shows comparisons to IIAB and Fail-Aware for random graphs with $N = 50, 75$, and 100.

As with UNOS graphs, results for MCTS and Greedy are quite similar, and both methods achieve larger $\Delta^{\texttt{MAX}}$ than Random, IIAB, and Fail-Aware. We make two observations: (1) Greedy appears to achieve larger $\Delta^{\texttt{MAX}}$ than MCTS in the single-stage setting, likely because of insufficient training time for MCTS; (2) in the multi-stage setting, MCTS performs *at least* as well as Greedy, and often better. Observation (2) is consistent with our experiments on UNOS graphs, and is somewhat surprising given that MCTS used less training time in these experiments. This suggests that MCTS may substantially improve over Greedy in the multi-stage setting; we leave further investigation to future work.

## D  Proofs for Section 2

In the proofs of Proposition 2.1 and Proposition 2.2 we consider a setting where all edges' pre-match rejections and post-match failures are i.i.d., where $P_R = 0.5$ is the pre-match rejection probability, $P_Q = 1.0$ is the post-match success probability if the edge is queried-and-accepted,

(a) Single-stage.  (b) Multi-stage.

Figure 4: Results for 30 random graphs with edge probability $p = 0.01$ and $N = 50$ vertices (top row), $N = 75$ (middle row), and $N = 100$ (bottom row). All experiments use the *Simple* edge distribution. In all plots, a solid line indicates median $\Delta^{\mathtt{MAX}}$ over all 30 random graphs, and shading is between the $10^{th}$ and $90^{th}$ percentiles; a dotted line indicates the baseline.

Figure 5: Exchange graph for Propositions 2.1 and 2.2. All edges have weight 1 except for edge $(E, B)$, which has weight 1.5.

479  and $P_N = 0.5$ is the success probability if $e$ is not queried. That is, queried edges have rejection
480  probability 0.5, accepted edges have zero failure probability, and non-queried edges have failure
481  probability 0.5.

## D.1 Proof of Proposition 2.1

483  (Proof by counterexample.) We provide an example where querying a single edge results in a *lower*
484  objective value in Problem 1 (i.e., final expected matching weight) than querying no edges–when
485  using the max-weight matching policy $M^{\mathtt{MAX}}(\cdot)$.

486  Consider the exchange graph in Figure 5; edge $(E, B)$ has weight 1.5, while all other edges have
487  weight 1. First we consider the objective due to querying no edges, $V^S(\mathbf{0})$. In this case, no edges
488  can be rejected pre-match, the max-weight matching includes cycle $(C, D, F)$ (expected weight
489  $3 \times (1/2)^3 = 3/8$) and cycle $(A, B)$ (expected weight $2 \times (1/2)^2 = 1/2$), with total expected
490  matching weight $7/8$. That is, $V^S(\mathbf{0}) = 7/8$.

491  Next consider the objective due to querying only edge $e_3 = (C, D)$, and let $\boldsymbol{q}'$ denote edge set
492  $\{e_3\}$. With probability $1/2$, $e_3$ is rejected and cycle $(B, C, E)$ is the max-weight matching – with

expected weight $3.5/8$. With probability $1/2$, $e_3$ is accepted and the max-weight matching includes cycles $(A, B)$ (with expected weight $1/2$) and $(C, D, F)$ (with expected weight $3/4$); this matching has total expected weight $5/4$. Thus, $V^S(\boldsymbol{q}) = 27/32 < 7/8 = V^S(\boldsymbol{0})$, which concludes the proof.

## D.2  Proof of Proposition 2.2

(Proof by counterexample.) We provide an example where the objective value in Problem 1 (i.e., final expected matching weight) is non-submodular–when using the max-weight matching policy $M^{\mathtt{MAX}}(\cdot)$. We use the same rejection and failure distribution as in the proof of Proposition 2.1.

Consider the exchange graph in Figure 5; edge $(E, B)$ has weight $1.5$, while all other edges have weight $1$. With some abuse of notation, we will denote by $V^S(\{e_a, \ldots, e_N\})$ the objective of Problem 1 due to edge set $\{e_a, \ldots, e_N\}$. Our counterexample for submodulartiy is that, for this graph,

$$V^S(X \cup \{e_1, e_2\}) + V^S(X) > V^S(X \cup \{e_1\}) + V^S(X \cup \{e_2\}),$$

with set $X \equiv \{e_3\}$. That is, the objective increase due to of querying *both* edges $e_1$ and $e_3$ is greater than the combined increase due to querying both edges separately. Next we explicitly calculate each of the above terms.

$V^S(X) = V^S(\{e_3\})$.  There are two cases to consider:

- $e_3$ is accepted, with probability $1/2$. The max-weight matching is cycles $(A, B)$ and $(C, D, F)$, with expected weight $(1/2 + 3/4)$,

- $e_3$ is rejected, with probability $1/2$. The max-weight matching is cycle $(B, C, E)$, with expected weight $3.5/8$.

Thus, $V^S(X) = (1/2)(1/2 + 3/4) + (1/2)(3.5/8) = 27/32$.

$V^S(X \cup \{e_1\}) = V^S(\{e_1, e_3\})$.  There are four cases to consider:

- $e_1$ and $e_3$ are accepted, with probability $1/4$. The max-weight matching is cycles $(A, B)$ and $(C, D, F)$, with expected weight $(1 + 3/8)$,

- $e_1$ is rejected and $e_3$ is accepted, with probability $1/4$. The max-weight matching is cycle $(B, C, E)$, with expected weight $3.5/8$.

- $e_1$ is accepted and $e_3$ is rejected, with probability $1/4$. The max-weight matching is cycle $(B, C, E)$, with expected weight $3.5/8$.

- $e_1$ and $e_3$ are rejected, with probability $1/4$. The max-weight matching is cycle $(B, C, E)$, with expected weight $3.5/8$.

Thus the objective is $V^S(X \cup \{e_3\}) = (1/4)(1 + 3/8) + (3/4)(3.5/8) = 43/64$.

$V^S(X \cup \{e_2\}) = V^S(\{e_2, e_3\})$.  There are three cases to consider

- $e_3$ is accepted, with probability $1/2$. The max-weight matching is cycles $(A, B)$ and $(C, D, F)$, with expected weight $(1/2 + 3/4)$,

- $e_3$ is rejected and $e_3$ is accepted, with probability $1/4$. The max-weight matching is cycle $(B, C, E)$, with expected weight $3.5/4$,

- $e_3$ and $e_2$ are rejected, with probability $1/4$. The max-weight matching is cycle $(A, B)$, with expected weight $1/2$.

Thus the objective is $V^S(X \cup \{e_2\}) = (1/2)(1/2 + 3/4) + (1/4)(3.5/4) + (1/4)(1/2) = 31/32$.

$V^S(X \cup \{e_1, e_2\}) = V^S(\{e_1, e_2, e_3\})$.  There are four cases to consider:

- $e_1$ and $e_3$ are accepted, with probability $1/4$. The max-weight matching is cycles $(A, B)$ and $(C, D, F)$, with expected weight $(1 + 3/4)$,

- $e_1$ is accepted and $e_2$ is rejected, with probability $1/4$ (the response from $e_3$ is irrelevant). The max-weight matching is $(A, B)$ and $(C, D, F)$, with expected weight $1 + 3/8$.

- $e_1$ is rejected and $e_2$ is accepted (the response from $e_3$ is irrelevant), with probability $1/4$. The max-weight matching is cycle $(B, C, E)$, with expected weight $3.5/4$.

- $e_1$ and $e_2$ are rejected (the response from $e_3$ is irrelevant), with probability $1/4$. The max-weight matching is cycle $(C, D, F)$, with expected weight $3/8$.

Thus the objective is $V^S(X \cup \{e_1, e_2\}) = (1/4)(1 + 3/4) + (1/4)(1 + 3/8) + (1/4)(3.5/4) + (1/4)(3/8) = 35/32$.

Finally, we have:

$$V^S(X \cup \{e_1, e_2\}) + V^S(X) = 35/32 + 27/32$$
$$= 1.9375$$

and

$$V^S(X \cup \{e_1\}) + V^S(X \cup \{e_2\}) = 43/64 + 31/32$$
$$= 1.640625$$

Therefore, $V^S(X \cup \{e_1, e_2\}) + V^S(X) > V^S(X \cup \{e_1\}) + V^S(X \cup \{e_2\})$, which concludes the proof.

### D.3 Proof of Proposition 2.3

For the proof of Proposition 2.3 we make one assumption about the distribution of edge rejections and failures: querying *additional* edges cannot increase the overall probability of rejection or failure for any edge.

**Assumption D.1.** *Let $q, r \in \{0, 1\}^{|E|}$ denote initial edge queries and responses. Let $q'$ be additional edges, such that $q + q' \in \{0, 1\}^{|E|}$ denotes an augmented edge set; let $r' \in \{0, 1\}^{|E|}$ denote the responses to edges $q'$ only. We assume that for any such $q$, $r$, and $q'$,*

$$\mathbb{E}\left[r + f \mid q, r\right] \geq \mathbb{E}\left[r + r' + f \mid q + q', r\right] .$$

Intuitively, Assumption D.1 excludes distributions where queries arbitrarily increase edge failure or rejection. For example, Assumption D.1 disallows the following distribution: suppose all edges are independent; all queried edges are accepted ($P(r_e = 1 \mid q) = 0$ for all $q$), all accepted edges have failure probability 0.5 ($P(f_e = 1 \mid q_e = 1, r_e = 0) = 0.5$), and all non-queried edges have failure probability 0.1 ($P(f_e = 1 \mid q_e = r_e = 0) = 0.1$). In this case, if an edge is not queried, then it has overall rejection or failure probability 0.1 (i.e., $\mathbb{E}[r_e + f_e \mid q, r] = 0.1$ with $q_e = 0$); if this edge is queried, then it has rejection or failure probability 0.5 (i.e., $\mathbb{E}[r_e + r'_e + f_e \mid q + q', r] = 0.5$ with $q'_e = 1$).

First we prove a handful of useful results.

**Definition D.2** (Edge Independence). *Two edges $e, e' \in E$ are independent if (a) their rejection distributions are conditionally independent, given whether or not they were queried:*

$$r_e \perp\!\!\!\perp r_{e'} \mid q_e \quad and \quad r_e \perp\!\!\!\perp r_{e'} \mid q_{e'}$$

*and (b) their failure distributions are conditionally independent, given whether or not they were queried and rejected:*

$$f_e \perp\!\!\!\perp f_{e'} \mid q_e, r_e \quad and \quad f_e \perp\!\!\!\perp f_{e'} \mid q_{e'}, r_{e'} .$$

**Lemma D.3.** *If all edges are independent, then additional edge queries cannot decrease expected post-match cycle and chain weights. Formally,*

$$\mathbb{E}\left[F(c, r + f) \mid q, r\right] \leq \mathbb{E}\left[F(c, r + r' + f) \mid q + q', r\right]$$

*for any $q, q' \in \{0, 1\}^{|E|}$ such that $q + q' \in \{0, 1\}^{|E|}$, for any $r \in \{0, 1\}^{|E|}$, and for all $c \in \mathcal{C}$.*

*Proof.* We address cycles and chains separately.

**Cycles.** Conditional on fixed $q$ and $r$, the expected weight of cycle $c = (e_1, \ldots, e_L)$ is expressed as

$$\mathbb{E}\left[F(c, r + f) \mid q, r\right] = \left(\sum_{e \in c} w_e\right) \mathbb{E}\left[\prod_{e \in c}(1 - r_e - f_e) \mid q, r\right]$$
$$= \left(\sum_{e \in c} w_e\right) \prod_{e \in c}(1 - \mathbb{E}\left[r_e + f_e \mid q, r\right])$$

where the second step is due to the fact that all $\boldsymbol{f}_e$ are independent. Similarly, for fixed $\boldsymbol{q}'$,

$$\mathbb{E}\left[F(c, \boldsymbol{r} + \boldsymbol{r}' + \boldsymbol{f}) \mid \boldsymbol{q} + \boldsymbol{q}', \boldsymbol{r}\right] = \left(\sum_{e \in c} w_e\right) \prod_{e \in c} \left(1 - \mathbb{E}\left[r_e + r'_e + \boldsymbol{f}_e \mid \boldsymbol{q} + \boldsymbol{q}', \boldsymbol{r}\right]\right) .$$

Due to Assumption D.1, the following inequality holds for all edges $e \in E$

$$\mathbb{E}\left[r_e + \boldsymbol{f}_e \mid \boldsymbol{q}, \boldsymbol{r}\right] \geq \mathbb{E}\left[r_e + r'_e + \boldsymbol{f}_e \mid \boldsymbol{q} + \boldsymbol{q}', \boldsymbol{r}\right] ,$$

and it follows that

$$\mathbb{E}\left[F(c, \boldsymbol{r} + \boldsymbol{f}) \mid \boldsymbol{q}, \boldsymbol{r}\right] \leq \mathbb{E}\left[F(c, \boldsymbol{r} + \boldsymbol{r}' + \boldsymbol{f}) \mid \boldsymbol{q} + \boldsymbol{q}', \boldsymbol{r}\right].$$

**Chains.** Similarly, the expected weight of chain $c = (e_1, \ldots, e_L)$ is expressed as

$$\mathbb{E}\left[F(c, \boldsymbol{r} + \boldsymbol{f}) \mid \boldsymbol{q}, \boldsymbol{r}\right] = \sum_{k=1}^{L} \left(\sum_{j=1}^{k} w_j\right) \mathbb{E}\left[\prod_{j=1}^{k} (1 - r_{e_j} - \boldsymbol{f}_{e_j}) \mid \boldsymbol{q}, \boldsymbol{r}\right]$$

$$= \sum_{k=1}^{L} \left(\sum_{j=1}^{k} w_j\right) \prod_{j=1}^{k} \left(1 - \mathbb{E}\left[r_{e_j} + \boldsymbol{f}_{e_j} \mid \boldsymbol{q}, \boldsymbol{r}\right]\right) ,$$

where the second step is due to the fact that $\boldsymbol{f}_e$ are independent. Similarly,

$$\mathbb{E}\left[F(c, \boldsymbol{r} + \boldsymbol{r}' + \boldsymbol{f}) \mid \boldsymbol{q} + \boldsymbol{q}', \boldsymbol{r}\right] = \sum_{k=1}^{L} \left(\sum_{j=1}^{k} w_j\right) \prod_{j=1}^{k} \left(1 - \mathbb{E}\left[r_{e_j} + r'_{e_j} + \boldsymbol{f}_{e_j} \mid \boldsymbol{q} + \boldsymbol{q}', \boldsymbol{r}\right]\right) .$$

as before, due to Assumption D.1 it follows that

$$\mathbb{E}\left[F(c, \boldsymbol{r} + \boldsymbol{f}) \mid \boldsymbol{q}, \boldsymbol{r}\right] \leq \mathbb{E}\left[F(c, \boldsymbol{r} + \boldsymbol{r}' + \boldsymbol{f}) \mid \boldsymbol{q} + \boldsymbol{q}', \boldsymbol{r}\right].$$

□

**Lemma D.4.** *With a failure-aware matching policy, and if all edges are independent, adding a single edge to any edge query set weakly improves the objective of Problem 1. Formally, for any $\boldsymbol{q}, \boldsymbol{q}' \in \{0, 1\}^{|E|}$ with $\boldsymbol{q} + \boldsymbol{q}' \in \{0, 1\}^{|E|}$ and $|\boldsymbol{q}'| = 1$, and $M(\boldsymbol{r}) \equiv M^{\text{FA}}(\boldsymbol{r})$,*

$$V^S(\boldsymbol{q}) \leq V^S(\boldsymbol{q} + \boldsymbol{q}')$$

*Proof.* The objective of Problem 1 for edge set $\boldsymbol{q}$ is expressed as

$$V^S(\boldsymbol{q}) = \mathop{\mathbb{E}}_{\boldsymbol{r}|\boldsymbol{q}} \left[ \mathop{\mathbb{E}}_{\boldsymbol{f}|\boldsymbol{q},\boldsymbol{r}} \left[ \sum_{c \in \mathcal{C}} M_c^{\text{FA}}(\boldsymbol{r}) F(c, \boldsymbol{r} + \boldsymbol{f}) \right] \right]$$

$$= \sum_{\boldsymbol{r} \in \{0,1\}^{|\boldsymbol{q}|}} P_{\boldsymbol{q}}(\boldsymbol{r}) \mathop{\mathbb{E}}_{\boldsymbol{f}|\boldsymbol{q},\boldsymbol{r}} \left[ \sum_{c \in \mathcal{C}} M_c^{\text{FA}}(\boldsymbol{r}) F(c, \boldsymbol{r} + \boldsymbol{f}) \right]$$

$$= \sum_{\boldsymbol{r} \in \{0,1\}^{|\boldsymbol{q}|}} P_{\boldsymbol{q}}(\boldsymbol{r}) \sum_{c \in \mathcal{C}} M_c^{\text{FA}}(\boldsymbol{r}) \mathop{\mathbb{E}}_{\boldsymbol{f}|\boldsymbol{q},\boldsymbol{r}} \left[ F(c, \boldsymbol{r} + \boldsymbol{f}) \right]$$

For edge set $\boldsymbol{q} + \boldsymbol{q}'$ we partition response variables into $\boldsymbol{r}, \boldsymbol{r}' \in \{0, 1\}^{|E|}$, where $r_e$ is the response variable for all edges $e \in \boldsymbol{q}$, and $r_e = 0$ for all other edges (including the edge in $\boldsymbol{q}'$). Similarly, $r'_e$ is the response variable for edge $\boldsymbol{q}'$, and $r'_e = 0$ for all other edges. The objective of $\boldsymbol{q} + \boldsymbol{q}'$ is expressed as

$$V^S(\boldsymbol{q} + \boldsymbol{q}') = \mathop{\mathbb{E}}_{\boldsymbol{r}, \boldsymbol{r}'|\boldsymbol{q}+\boldsymbol{q}'} \left[ \mathop{\mathbb{E}}_{\boldsymbol{f}|\boldsymbol{q}+\boldsymbol{q}',\boldsymbol{r}+\boldsymbol{r}'} \left[ \sum_{c \in \mathcal{C}} M_c^{\text{FA}}(\boldsymbol{r} + \boldsymbol{r}') F(c, \boldsymbol{r} + \boldsymbol{r}' + \boldsymbol{f}) \right] \right]$$

$$= \sum_{\boldsymbol{r} \in \{0,1\}^{|\boldsymbol{q}|}} P_{\boldsymbol{q}+\boldsymbol{q}'}(\boldsymbol{r}) \mathop{\mathbb{E}}_{\boldsymbol{r}'|\boldsymbol{q}+\boldsymbol{q}'} \left[ \mathop{\mathbb{E}}_{\boldsymbol{f}|\boldsymbol{q}+\boldsymbol{q}',\boldsymbol{r}+\boldsymbol{r}'} \left[ \sum_{c \in \mathcal{C}} M_c^{\text{FA}}(\boldsymbol{r} + \boldsymbol{r}')^\top F(c, \boldsymbol{r} + \boldsymbol{r}' + \boldsymbol{f}) \right] \right]$$

$$= \sum_{\boldsymbol{r} \in \{0,1\}^{|\boldsymbol{q}|}} P_{\boldsymbol{q}}(\boldsymbol{r}) \mathop{\mathbb{E}}_{\boldsymbol{r}'|\boldsymbol{q}+\boldsymbol{q}'} \left[ \mathop{\mathbb{E}}_{\boldsymbol{f}|\boldsymbol{q}+\boldsymbol{q}',\boldsymbol{r}+\boldsymbol{r}'} \left[ \sum_{c \in \mathcal{C}} M_c^{\text{FA}}(\boldsymbol{r} + \boldsymbol{r}') F(c, \boldsymbol{r} + \boldsymbol{r}' + \boldsymbol{f}) \right] \right],$$

where in the final line we replace $P_{q+q'}(r)$ with $P_q(r)$, because each $r_e$ is conditionally independent, given $q_e$.

Next, by definition

$$\mathop{\mathbb{E}}_{f|q+q',r+r'}\left[\sum_{c\in\mathcal{C}} M_c^{\text{FA}}(r+r')F(c,r+r'+f)\right] \geq \mathop{\mathbb{E}}_{f|q+q',r+r'}\left[\sum_{c\in\mathcal{C}} x_c F(c,r+r'+f)\right] \quad \forall x\in\mathcal{M}.$$

That is, $M^{FA}$ is guaranteed to maximize this expectation, and thus

$$V^S(q+q') \geq \sum_{r\in\{0,1\}^{|q|}} P_q(r) \mathop{\mathbb{E}}_{r'|q+q'}\left[\mathop{\mathbb{E}}_{f|q+q',r+r'}\left[\sum_{c\in\mathcal{C}} M_c^{\text{FA}}(r)F(c,r+r'+f)\right]\right] \qquad \text{(B)}$$

$$= \sum_{r\in\{0,1\}^{|q|}} P_q(r) \sum_{c\in\mathcal{C}} M_v^{\text{FA}}(r) \mathop{\mathbb{E}}_{r'|q+q'}\left[\mathop{\mathbb{E}}_{f|q+q',r+r'}\left[F(c,r+r'+f)\right]\right] \qquad \text{(C)}$$

Finally, combining (B) and (C) with Lemma D.3, the following inequality holds

$$V^S(q) \leq V^S(q+q').$$

$\square$

Using the above lemmas, the proof of Proposition 2.3 is straightforward:

**Proposition 2.3**   *With a failure-aware matching policy, if all edges are independent, the objective of Problem 1 is monotonic in the set of queried edges.*

*Proof.* Let $q',q''\in\mathcal{E}$ be two edge sets such that $q'\subseteq q''$. It remains to show that, with matching policy $M(r)\equiv M^{\text{FA}}(r)$,

$$V^S(q'') \leq V^S(q').$$

First note that because $\mathcal{E}$ is a matroid, there is a sequence of edges $(q^{e_1},\ldots,q^{e_L})$ (with each $|q^{e_i}|=1$) such that $q''+q^{e_1}+\cdots+q^{e_L}=q'$. Due to Lemma D.4, the following sequence of inequalities hold:

$$V(q'') \leq V(q''+q^{e_1})$$
$$\leq V(q''+q^{e_1}+q^{e_2})$$
$$\cdots$$
$$\leq V(q''+q^{e_1}+\cdots+q^{e_L})$$
$$= V(q')$$

which concludes the proof. $\square$

# E   Algorithm Descriptions

Here we describe more explicitly the algorithms for `Greedy` and `MCTS`, for both the single-stage and multi-stage settings.

## E.1   UCB Value Estimates for `MCTS`

Both the single- and multi-stage versions of `MCTS` use the method of [18] to select the next child node to explore. The formula used to estimate a node's UCB value is

$$\frac{\frac{U}{N}-V^{min}}{V^{max}-V^{min}} + \sqrt{N^P/N}$$

where $U$ is the "UCB value estimate" calculated by `MCTS`, $N$ is the number of visits to the node, $N^P$ is the number of visits to the node's parent, and $V^{max}$ and $V^{min}$ are the largest and smallest *node values* encountered during search. In single-stage `MCTS`, all nodes have both a *node value* (the objective value of Problem 1) and a UCB value estimate; as described below, in multi-stage `MCTS` only query nodes have a UCB value estimate, and only leaf nodes have a *node value* (expected matched weight, after observing responses from all queried edges).

---

**ALGORITHM 3:** `Greedy`: Greedy Search Heuristic for Single-Stage Edge Selection

---

(input) $\mathcal{E}$: legal edge sets

$\boldsymbol{q}^R \leftarrow \boldsymbol{0}$     the root node (no edges)
$V^* \leftarrow$ objective value of $\boldsymbol{q}^R$ Problem 1
**while** $\boldsymbol{q}^R$ *has children* **do**
    $\boldsymbol{q}' \leftarrow$ child node of $\boldsymbol{q}^R$ with maximal objective value in Problem 1
    $\boldsymbol{q}^R \leftarrow \boldsymbol{q}'$
**return** $\boldsymbol{q}^R$

---

### E.2   Greedy Single-Stage Edge Selection

Algorithm 3 gives a pseudocode description of `Greedy` for the single-stage setting.

### E.3   Multi-Stage Edge Selection

In the following sections we describe multi-stage versions of `MCTS` and `Greedy`. Unlike in the single-stage setting, these algorithms take as input a set of previously-queried edges $\boldsymbol{q} \in \{0,1\}^{|E|}$ and a corresponding set of observed rejections $\boldsymbol{r} \in \{0,1\}^{|E|}$; they output the *next* edge to query.

**Multi-Stage** `MCTS`.   The multi-stage search tree is somewhat more complicated than in the single-stage setting, as each node in the search tree corresponds to both a set of queried edges and a set of observed rejections. For this purpose we use two types of nodes: *outcome* nodes, and *query* nodes. Outcome nodes consist of previously-queried edges $\boldsymbol{q}$ and previously-observed rejections $\boldsymbol{r}$, and are represented by tuple $(\boldsymbol{q}, \boldsymbol{r})$. (The root of the search tree corresponds to *no* queries or observed rejections, $(\boldsymbol{0}, \boldsymbol{0})$.) The children of an outcome node are *query* nodes, represented by the next edge to query from the parent (outcome), represented by tuple $(\boldsymbol{q}, \boldsymbol{r}, e)$. Each outcome node has one child for every edge that has not yet been queried:

$$C^O(\boldsymbol{q}, \boldsymbol{r}) \equiv \{(\boldsymbol{q}, \boldsymbol{r}, e) \mid \forall e \in E : \boldsymbol{q} + \boldsymbol{u}^e \in \mathcal{E}\}$$

where $\boldsymbol{u}^e$ is the unit vector for element $e$ ($\boldsymbol{u}_i^e = 0$ for all $i \neq e$, and $\boldsymbol{u}_e^e = 1$). Each query node has exactly two children: one where the queried edge is accepted, and one where the queried edge is rejected,

$$C^Q(\boldsymbol{q}, \boldsymbol{r}, e) \equiv \{(\boldsymbol{q} + \boldsymbol{u}^e, \boldsymbol{r}), (\boldsymbol{q} + \boldsymbol{u}^e, \boldsymbol{r} + \boldsymbol{u}^e)\}$$

As before, the *level* of a node refers to the number of queried edges: $|\boldsymbol{q}|$ for outcome nodes, and $|\boldsymbol{q}| + 1$ for query nodes.

As before we refer to nodes with no children as leaf nodes; note that only outcome nodes are leaf nodes. Unlike the single-stage version of `MCTS`, in the multi-stage setting we only consider the value of leaf nodes[6]. The value of a leaf (outcome) is

$$V^O(\boldsymbol{q}, \boldsymbol{r}) \equiv W(M(\boldsymbol{r}); \boldsymbol{q}, \boldsymbol{r}),$$

where as before $M(\boldsymbol{r})$ denotes the matching policy, and $W(\boldsymbol{x}; \boldsymbol{q}, \boldsymbol{r})$ denotes the expected matching weight of $\boldsymbol{x}$, subject to $\boldsymbol{q}$ and $\boldsymbol{r}$. The value of leaf outcome nodes is used to by `QSample` and `OSample` to guide multi-stage `MCTS`.

605 Algorithm 4 describes the multi-stage version of MCTS, taking previously-queried edges and
606 observed responses as input. This algorithm initializes the value estimate $U[\cdot]$ and number of visits
607 $N[\cdot]$ for query nodes in the next $L$ levels–these quantities are used in the UCB calculation.

---

**ALGORITHM 4:** Multi-Stage MCTS

---

(input) $\mathcal{E}$: legal edge sets
(input) $K$: maximum size of any legal edge set
(input) $T$: time limit
(input) $L$: number of look-ahead levels
(input) $\boldsymbol{q}^R$: previously-queried edges
(input) $\boldsymbol{r}^R$: previously-observed rejections

608
$M \leftarrow \min\{N + L, K\}$
$Q \leftarrow$ all query nodes which are descendants of $(\boldsymbol{q}^R, \boldsymbol{r}^R)$, up to level $M$
$U[(\boldsymbol{q}, \boldsymbol{r}, e)] \leftarrow 0 \; \forall (\boldsymbol{q}, \boldsymbol{r}, e) \in Q$    UCB value estimate
$N[(\boldsymbol{q}, \boldsymbol{r}, e)] \leftarrow 0 \; \forall \boldsymbol{q} \in Q$    number of visits
**while** *less than time $T$ has passed* **do**
      QSample($\boldsymbol{q}^R$, $\boldsymbol{r}^R$ $M$)
$(\boldsymbol{q}^R, \boldsymbol{r}^R, e^*) \leftarrow$ child node of $(\boldsymbol{q}^R, \boldsymbol{r}^R)$ with the greatest UCB estimate
**return** $e^*$

---

**ALGORITHM 5:** QSample: Function for sampling query nodes in multi-stage MCTS

---

(input) $(\boldsymbol{q}, \boldsymbol{r})$: outcome node
(input) $M$: maximum level to sample from

**if** $(\boldsymbol{q}, \boldsymbol{r})$ *has no children* **then**
      **return** $V^O(\boldsymbol{q}, \boldsymbol{r})$    (return the value of this outcome node)
609 **if** $(\boldsymbol{q}, \boldsymbol{r})$ *has children* **then**
      **if** $|\boldsymbol{q}| < M - 1$ **then**
            $(\boldsymbol{q}, \boldsymbol{r}, e') \leftarrow$ child node of $(\boldsymbol{q}, \boldsymbol{r})$ with the greatest UCB estimate
            OSample($\boldsymbol{q}, \boldsymbol{r}, e$)
      **else**
            $(\boldsymbol{q}', \boldsymbol{r}') \leftarrow$ random leaf node, descendant from $(\boldsymbol{q}, \boldsymbol{r})$
            **return** $V^O(\boldsymbol{q}', \boldsymbol{r}')$

---

**ALGORITHM 6:** OSample: Function for sampling outcome nodes in multi-stage MCTS

---

(input) $(\boldsymbol{q}, \boldsymbol{r}, e)$: query node

610
$N[(\boldsymbol{q}, \boldsymbol{r}, e)] \leftarrow N[(\boldsymbol{q}, \boldsymbol{r}, e)] + 1$
$\boldsymbol{q}' \leftarrow \boldsymbol{q} + \boldsymbol{u}^e$ (new query vector with edge $e$ added)
$Z \leftarrow$ randomly sample a response to edge $e$ (0 if accept, 1 if reject)
$\boldsymbol{r}' \leftarrow \boldsymbol{r} + Z\boldsymbol{u}^e$    (updated rejection vector)
$U[(\boldsymbol{q}, \boldsymbol{r}, e)] \leftarrow U[(\boldsymbol{q}, \boldsymbol{r}, e)] + \text{QSample}(\boldsymbol{q}', \boldsymbol{r}')$

---

611 Algorithm 5 (QSample) samples query nodes from an outcome node, while Algorithm 6 (OSample)
612 samples outcome nodes from a query node (and updates the query node's UCB value estimate).

613 **Multi-Stage Greedy.**    Algorithm 7 gives a pseudocode description of the multi-stage version of
614 Greedy. This search heuristic returns the next edge to query with the highest expected final matching
615 weight, *ignoring all future queries*. In other words, this approach treats every edge as the *last* edge;
616 one might call this heuristic "myopic" as well as greedy.

**ALGORITHM 7:** Greedy Heuristic for Multi-Stage Edge Selection

---

(input) $\mathcal{E}$: legal edge sets
(input) $\boldsymbol{q}$: previously-queried edges
(input) $\boldsymbol{r}$: previously-observed rejections

$e^* \leftarrow \emptyset \ V^* \leftarrow 0$
**for** *all $\boldsymbol{q}'$ in $\boldsymbol{q}$'s children* **do**
    $e' \leftarrow$ the new edge queried in child node $\boldsymbol{q}'$
    $\boldsymbol{r}^A \leftarrow \boldsymbol{r}$
    $\boldsymbol{r}^R \leftarrow \boldsymbol{r}$
    $\boldsymbol{r}^A_{e'} \leftarrow 0$   (response scenario where $e'$ is accepted, and $\boldsymbol{r}_{e'} = 0$)
    $\boldsymbol{r}^R_{e'} \leftarrow 1$   (response scenario where $e'$ is rejected, and $\boldsymbol{r}_{e'} = 1$)
    $p^A \leftarrow$ probability that $e$ is accepted, conditional on previous responses
    $p^R \leftarrow$ probability that $e$ is rejected, conditional on previous responses
    $V' \leftarrow p^A \cdot W(M(\boldsymbol{r}^A); \boldsymbol{q}', \boldsymbol{r}^A) + p^R W(M(\boldsymbol{r}^R); \boldsymbol{q}', \boldsymbol{q}^R)$   (value of querying edge $e'$)
    **if** $V' > V^*$ **then**
        $\boldsymbol{e}^* \leftarrow e'$
        $V^* \leftarrow V'$
**return** $\boldsymbol{e}^*$

---

## Footnotes

[6]This decision was made in part because initial results indicate that edge selection is essentially monotonic.