[Reviews · NeurIPS 2020]

Review 1

Summary and Contributions: This paper studies a kidney exchange problem. Kidney exchanges can be represented by a directed graph. Each edge is considered as one potential kidney exchange, and the goal is to choose a subset of edges (i.e., a matching policy), which maximizes the expected number of successful exchanges. The key element in the model is the ability of the decision-maker to perform a pre-screening (i.e., query) before constructing a matching. After a query, a binary label is assigned to each queried edge: Accepted or Rejected. If the label of a queried edge is “rejected,” it will not be selected in matching phase. On the other hand, an edge with “accepted” label may or may not be selected by matching policy. We should keep in mind an “accepted” label does not guarantee a successful exchange. The main contribution of this paper is developing an algorithm for selecting query edges. Finding optimal pre-match queries is an NP-hard problem. However, this paper suggests two algorithms (MCTS and Greedy) to find a suboptimal pre-match query.

Strengths: 1. The first contribution is the model itself. This model considers the possibility of performing pre-screening before constructing a matching. A queried edge can be labeled as accepted or rejected. However, an edge with "accepted" label does not guarantee a successful exchange. This proposed kidney exchange model with pre-screening is new. 2. Finding the optimal query is an NP-hard problem. However, this paper proposes an algorithm to find a suboptimal query strategy that helps the decision-maker construct better matching. Based on table 1, the proposed algorithm finds the optimal query for at least 90 of the 100 graphs.

Weaknesses: I have the following concerns about the paper: 1. This paper introduces an algorithm to find a suboptimal query strategy, and it does not provide an algorithm to find the optimal matching. For finding the best query strategy, we need to know the optimal matching policy as a function of the query edges. This paper assumes that after selecting a specific query, the optimal matching and the maximum utility are available to the decision-maker. More precisely, this paper assumes that $V^S(q)$ is available to the decision-maker. However, we need to find the optimal matching policy for evaluating $V^S(q)$. Finding the optimal matching is much more difficult and important problem. Of course, after evaluating $V^S(q)$, we can search through possible queries to find a (sub)optimal query. Even if the decision-maker does not want to perform pre-screening, he has to find the optimal matching, and this paper does not provide an algorithm for that. 2. The experiment is unfair. Just consider the definition of $\Delta^{max}$. $\Delta^{max}$ tells us how much improvement the proposed algorithm can make as compared to a scenario without query. Of course, a scenario with a query can achieve higher utility and outperform a matching without pre-screening/query. Therefore, figure 2 is not informative. I found the left part of Table 1 very informative, though. It shows that the Greedy algorithm can find the optimal query in 90 graphs out of 100 graphs. It implies that the Greedy algorithm is a powerful method. 3. There is no complexity analysis of the proposed algorithms. That would be great if the authors provide the run time of the proposed algorithms. To find an optimal query, we have to find the optimal matching associated with each query. Therefore, I feel finding a (sub)optimal query would be very expensive. By the right part of table 1, a random query can improve the objective function of the problem (1) by 50% as compared to a scenario without query. Given the fact that choosing a random query does not have any computational cost, a random query may be a better choice as compared to the Greedy or MCTS algorithm. 4. This paper compares its results with [10]. Since [10] considers a different model, the authors should mention how the algorithm of [10] can solve optimization problem (1). 5. Since [15] does not use a query to find the optimal matching, it is not fair to compare MCTS/Greedy algorithm with the algorithm of [15]. UPDATE: The authors have addressed my concecerns with the experiment section. Moreover, even though their proposed algorithms are simple and are commonly used in the literature, they show that they work on real data and in kidney exchange problem. Therefore, I would like to increase the score.

Correctness: Claims.propositions are correct. As I mentioned in the previous part, this paper proposes an algorithm to find the optimal query. However, it does not provide an algorithm to find the optimal matching. To find the optimal query, we need to know the optimal matching and maximum utility/objective function for any given query. Any query, gives us a rejection vector $r$ and the rejection vector affects the optimal matching strategies $M^{max}(r)$ and $M^{FA}(r)$. Therefore, finding an algorithm for an optimal query without having an algorithm to find $M^{max}(r)$ or $M^{FA}(r)$ does not seem right.

Clarity: Yes, it is. It was easy to read and understand the model, main idea and emprical methodology.

Relation to Prior Work: Yes, it is. As I mentioned before, the model itself is new and makes the paper different from existing literature.

Reproducibility: Yes

Additional Feedback: That would be intresting if the authros try to find an algorithm which finds the optimal matching given a specific query. Then, the algorithm can be used in proposed MCTS or Greedy algorithm to find the best query.


Review 2

Summary and Contributions: This paper points out the unrealistic assumption of prior work in the kidney exchange problem and proposes a multi-stage optimization problem with both the pre- and post-match phases. The authors also demonstrate that this problem set is a mathematically challenging combinatorial problem, and propose a greedy heuristic and Monte Carlo tree search method to solve this problem. Finally, through synthetic data and real data, they experimentally confirm that their pre-screening has a significant effect on performance.

Strengths: The authors suggest a new research direction in the kidney exchange problem and show that the problem they propose is theoretically and empirically meaningful. Also, since their problem setting considers the structure of the actual kidney exchange at the application level, this paper seems to be a good initial study for those who study this field.

Weaknesses: (1) Even considering this is an initial study through a new problem formulation, the algorithmic contribution proposed in the paper is insufficient. They experimentally show that the simple greedy method and the Monte Carlo tree search method are effective in this NP-hard problem, but there is no mathematical discussion with this claim regarding the characteristics of the data. Alternatively, suggesting an algorithm modification that considers the characteristics of the problem can also be a way for algorithmic contributions (2) If simple greedy or Monte Cralot tree search in the kidney exchange problem ensures sufficient performance, formulating this problem as an edge query problem does not seem to be of much value. The problem of estimating the probability distribution or graph assuming given in this problem seems more realistic and important.

Correctness: The claims and methods are clear and correct

Clarity: (1) The abstract part is too long. The detailed explanations at the beginning of the abstract seem to need to be reduced. (2) In 4.1 and 4.2, it is inconvenient to read because the experimental setting and the experimental result are mixed within one paragraph. (3) Multistage and multi-stage are used interchangeably.

Relation to Prior Work: This paper introduces a sufficient level of prior work, and the authors compare their contributions with existing research well.

Reproducibility: Yes

Additional Feedback:


Review 3

Summary and Contributions: This paper proposed the policy-constrained edge query model for kidney exchange problem, where a decision-maker selects a set of potential edges to pre-screen and then constructs a final packing using a fixed algorithm. This model generalizes existing models in the literature, as edge failure probabilities depend on whether or not the edge is pre-screened. The authors proved that the edge query problem is non-monotonic and non-submodular in the set of queried edges when the decision-maker uses a max-weight packing policy. Experiments are conducted on both simulated and real exchange data from the United Network for Organ Sharing (UNOS), showing that the proposed methods substantially outperform prior approaches.

Strengths: This paper considers the problem of kidney exchange, which is a practical real-world problem. The results of the experiments showing the performance of the algorithm are encouraging and the contribution has significant broader impact. Besides formulating the problem, the authors also prove the important properties (non-monotonic and non-submodular) of the problem and proposed baseline solutions with experiments, ensuring the soundness of the claims. The problem definition, proofs of propositions and algorithm description in section 2, 3 and appendix are written in a detailed and clear way. The writing of this paper is friendly to readers who are not familiar with the field.

Weaknesses: The proposed solutions to the problem (Monte Carlo Tree Search and the Single Stage Greedy Algorithm) have been used in other applications. The authors are encouraged to clarify special tweaks of these algorithms to make them work on the specific application.

Correctness: Yes.

Clarity: The writing of this paper is very clear and friendly to readers who are not familiar with the field. In section 2, the problem definition and proofs of propositions are provided, with detailed definitions and explanations of important concepts (matching policy, single or multi-stage setting, etc). Proofs of important properties are provided. In section 3, the algorithm description is provided in a detailed and clear way. The abstract and introduction of this paper are brief and clear, giving a good overview of the main idea.

Relation to Prior Work: The relation to prior work is provided clearly in the introduction section. Existing matching algorithms aiming to mitigate transplant failures usually require modifying fielded matching algorithms, which in many cases would require changing the law or policy. Pre-screening potential transplants can avoid failures without modifying the matching algorithm, but it is costly as it requires scarce time and resources. To overcome these problems, the multistage stochastic optimization problem formulation is proposed.

Reproducibility: Yes

Additional Feedback:

[Author Response · NeurIPS 2020]

We thank all reviewers (**R1**, **R2**, **R4**) for helpful and insightful comments!

**Matching Algorithm (R1).** One major concern for **R1** is that we do not provide an algorithm to find
the optimal matching (an NP-hard problem). We use the state-of-the-art method, "PICEF", of [15]
(Dickerson et al.) to find optimal matchings. PICEF takes only fractions of a second for realistic
exchange graphs (including those in our experiment). Our proposed edge-query methods (`Greedy`
and `MCTS`) both use PICEF as a low- or no-cost matching oracle. While this is a good assumption on
realistic kidney exchange graphs, this is not true on all instances because that problem is NP-hard.

**Benchmark Methods, and $\Delta^{\texttt{MAX}}$ (R1, R2).** In short, $\Delta^{\texttt{MAX}}$ compares edge-query methods against
the benchmark of max-weight matching (without edge queries). To clarify, we compare our methods
against two benchmark policies: the max-weight matching (policy $M^{\texttt{MAX}}(\cdot)$, which does not consider
edge uncertainty), and the stochastic matching policy of [16] (Dickerson et al.) (which maximizes
expected matching weight). As the reviewers point out, these policies do not query edges before
constructing the final matching—which is the present reality in all fielded kidney exchanges. We
compare against these policies to illustrate the improvement due to querying edges; we agree that it is
unfair to compare an edge-query method to policy $M^{\texttt{MAX}}(\cdot)$ (which does not query edges). In other
words, we compare against these benchmark policies to (a) show the potential gain from querying
edges, and (b) compare different edge-query policies to each other. In our final submission we will
clarify which policies are benchmarks, and we will elaborate on our reasons for including them.

**Random Performs Well (Table 1) (R1).** We believe this comment is in reference to the column
labelled $P_{90}$ in Table 1. These number are the 90th percentile of $\Delta^{\texttt{MAX}}$ over all exchange graphs,
meaning that Random gives this improvement (about $50\%$) in only $10\%$ of all graphs. On the other
hand, by looking at the column labelled $P_{10}$ (the 10th percentile) `Greedy` and `MCTS` achieve a similar
improvement ($40\%$) for $90\%$ of all graphs. In summary: Random works well on about $10\%$ of all
graphs, while `Greedy` and `MCTS` perform well on *most* graphs.

**Complexity & Runtime Analysis (R1).** Thanks for bringing this up. We will include the following:
Our methods rely on an "oracle" to solve the NP-hard kidney exchange matching problem. We can
report the *number of calls* to this oracle for each method as a measure of complexity. Both benchmark
methods (max-weight matching and failure-aware [16]) as well as `IIAB` (Blum et al. [10]) use exactly
one oracle call; i.e., they are $O(1)$. Both `Greedy` and `MCTS` use a fixed number of samples ($M$) to
evaluate the objective of an edge set. `Greedy` evaluates the objective of an edge set exactly $\Gamma$ times;
thus, `Greedy` is $O(M \cdot \Gamma)$. Finally, `MCTS` can in theory visit all potential edge sets of size at most $\Gamma$
(i.e., an exhaustive search), which is $O(M \cdot \sum_{\gamma=1}^{\Gamma} \binom{|E|}{\gamma})$. Since this version of `MCTS` is intractable in
both runtime and memory, we impose reasonable limits (see § 3 of our paper).

**Focusing on Edge Selection vs. Edge Failure Probabilities (R2).** We agree that our results suggest
that the edge query problem is easy in practice: perhaps our most important finding is that *in theory*
edge selection is hard (both non-monotonic and non-submodular, as well as NP-hard), while *in*
*practice* this problem is easy (a greedy heuristic is nearly optimal). Importantly, we see our paper as
*complementary* to ongoing research on edge failure distributions. We are aware of that research, and
it informs our own investigations. In summary, several research questions address edge-existence
uncertainty in kidney exchange: one is *Which edges fail and why?* (**R2**'s suggestion); another is,
*Given an edge-failure model, how do we improve the outcome of kidney exchange?* (the question we
address in our paper). These are complementary questions that both improve kidney exchange.

**Algorithmic contribution (R2).** We focus on motivating and formalizing the problem of selecting
edges to query–and we prove that this is not a theoretically easy task via Prop. 2.1 and 2.2. Yet, we
demonstrate that simple algorithms work well on real data. We strongly believe that our contributions
are novel, and will serve as a foundation for future work. Indeed, we are the first (to our knowledge)
to formalize this edge selection problem, either in the single-stage or multi-stage setting. Further, we
are the first to formalize and test simple algorithms for this setting; the only similar work is that of
Blum et al. [10], which considers a less-realistic setting (discussed in §1 of our paper).

**Clarity of Exposition (R1, R2, R4).** We thank all reviewers for clarity suggestions. In particular, we
will clarify our experimental sections (4.1 and 4.2) to better separate our experimental design from
our results. Finally, we will clarify details of our algorithms in the main paper. Currently the appendix
describes all algorithms in detail; we will move some important details to the paper, including the
UCB bounds used by our `MCTS` method and a discussion of the main hyperparameters of `MCTS` and
the values we chose. This will be possible because an extra page is allowed in the final version.



[Meta-Review · NeurIPS 2020]

This paper presents an interesting, and socially valuable, real-world application report to the NeurIPS community. It provides a good balance of theoretical exploration and practical analysis, and the performance improvement on real-world data is compelling. In assessing "application reports" from all domains for suitability for NeurIPS, we often ask "what would the general ML practitioner learn from this application", and here I believe the learning about ML is relatively small, however I do believe that exposure to the problem has value to the community, so I lean on the side of acceptance.